# Long Non-Coding RNA Function in CD4^+^ T Cells: What We Know and What Next?

**DOI:** 10.3390/ncrna5030043

**Published:** 2019-07-12

**Authors:** Katie A. West, Dimitris Lagos

**Affiliations:** 1York Biomedical Research Institute, University of York, Wentworth Way, York YO10 5DD, UK; 2Department of Biology, University of York, Wentworth Way, York YO10 5DD, UK; 3Hull York Medical School, University of York, Wentworth Way, York YO10 5DD, UK

**Keywords:** long non-coding RNA, adaptive immunity, Th cells, inflammation, infection

## Abstract

The non-coding genome has previously been regarded as “junk” DNA; however, emerging evidence suggests that the non-coding genome accounts for some of the greater biological complexity observed in mammals. Research into long non-coding RNAs (lncRNAs) has gathered speed in recent years, and a growing body of evidence has implicated lncRNAs in a vast range of cellular functions including gene regulation, chromosome organisation and splicing. T helper cells offer an ideal platform for the study of lncRNAs given they function as part of a complex cellular network and undergo remarkable and finely regulated gene expression changes upon antigenic stimulation. Using various knock down and RNA interaction studies several lncRNAs have been shown to be crucial for T helper cell differentiation, activation and function. Given that RNA targeting therapeutics are rapidly gaining attention, further understanding the mechanistic role of lncRNAs in a T helper context is an exciting area of research, as it may unearth a wide range of new candidate targets for treatment of CD4^+^ mediated pathologies.

## 1. The Non-Coding Transcriptome

Mammalian cells are capable of transcribing approximately 80% of their genome [1,2]. Historically, protein coding genes have been the focus of extensive study; however the fully spliced, protein-coding transcript isoforms of these genes comprise just above 2% of the genome [3]. The remaining untranslated genes generate non-coding RNAs (ncRNA), which previously have been regarded as transcriptional noise. In more recent years, some ncRNAs have been shown to play a vital role in gene regulation, nuclear organisation and are crucially implicated in multiple diseases [4,5]. Further understanding the mechanistic and biological relevance of non-coding RNA in disease is now under extensive investigation, and is an exciting challenge that faces molecular cell biology. In combination with advances in RNA-targeting therapeutics, the study of ncRNAs could unveil a multitude of potential drug targets.

ncRNAs are broadly classified into groups based on their post transcriptional length. Those which are less than 200 bp are known as small non-coding RNAs; examples include microRNA (miRNA), transfer ribonucleic acid (tRNA) and small nuclear RNA (snRNA). Conversely, transcripts longer than 200 bp are known as long non-coding RNA (lncRNA) [4]. Here, we will focus on the role and regulation of lncRNAs.

The estimated number of mammalian lncRNA genes ranges from 20,000 to 100,000 [6,7]. Many lncRNAs are transcribed by RNA polymerase II, consequently some of the transcripts have a 5′ cap and poly A tail. These transcripts are localised in various cellular compartments such as the nucleus or cytoplasm, are typically expressed at low copy numbers and can be poorly evolutionarily conserved. Given the archetypal low expression and poor sequence conservation of lncRNAs this draws the cellular relevance of some lncRNAs into question. However, many lncRNAs are expressed in a highly cell type- and tissue-specific manner, strongly suggesting lncRNAs may play a cell specific role. Although the exact function or lack thereof is not known for all of these lncRNA genes, those which have been studied can be classified on their mode of action as *cis* and/or *trans* acting lncRNA, through modular interactions with RNA, DNA and proteins [5,8]. LncRNAs which act in *trans* exert their function at a different site to which they were transcribed. In contrast, *cis* acting lncRNAs function by interacting with genes neighbouring their site of transcription and can help localise epigenetic modifiers to these locations [9]. However, there are many exceptions and nuances to this rule. An example of this is the lncRNA functional intergenic repeating RNA element (FIRRE). FIRRE originates from the X chromosome and interacts with the nuclear-matrix factor hnRNPU to contact regions located far away on the same chromosome [10]. In cases such as this, it is particularly difficult to assign an *“in cis*” or “*in trans*” mode of action. Nevertheless, it is clear lncRNAs play a role in a wide range of cellular functions both locally and distally.

Remarkably, in eukaryotes, as the complexity of an organism increases, the ratio of non-coding to protein coding DNA rises. Approximately 98.5% of the human genome is comprised of non-coding DNA, contrasting to 25–50% in simple eukaryotes, and >50% in complex fungi and plants [11]. This suggests that non-coding transcripts may account for some of the greater complexity seen in mammals.

The immune system contains some of the most complex cellular networks observed in mammals. It provides essential protection against an unpredictable and vast number of pathogens. Cells derived from both the innate and adaptive arms of the immune system undergo complex development, translocate throughout the body, and must act in a highly coordinated manner to remove threats. The exquisite specificity of this response results in an extremely sophisticated defence mechanism enabling protection against infection and reinfection throughout life. Thus, mammalian immunity is an ideal platform for the study of lncRNAs.

## 2. CD4^+^ T Cells

CD4^+^ T cells, also known as T helper cells, represent a crucial portion of the immune mechanism maintaining host protection. Initially, two distinct populations of CD4^+^ T cells were identified, each with unique cytokine expression profiles: Th1 cells, which secrete interferon gamma (IFNγ), and Th2 cells, which have interleukin 4 (IL-4) as their signature cytokine [12,13]. New subsets of CD4^+^ T cells have been discovered, each with a unique cytokine profile (Figure 1). These are broadly separated into two subsets: effector CD4^+^ T cells, which are key in host defence against pathogens (Th1, Th2, Th9, Th17), and a regulatory CD4^+^ subset, which dampen the immune response to aid prevention of autoimmunity (Tr1, Th3, nTreg, iTreg) [14]. CD4^+^ T cells undergo remarkable and finely regulated gene expression changes upon antigenic stimulation, which can be studied and manipulated in both in vivo and in vitro models.

Interestingly, RNA-seq data gathered from 42 subsets of murine thymocytes and CD4^+^ T cells at various time points of development identified 1524 lncRNA-expressing genomic regions (clusters), some of which encode more than one lncRNA. Between 48–57% of the identified lncRNAs were indicated to be cell type specific, contrasting with 6–8% of mRNAs [15]. The number of lncRNA clusters in each T cell subset was between 154 and 353. Many of the detected lncRNAs are genomic neighbours of important immunoregulatory genes and contain binding sites for CD4^+^ T cell transcription factors such as signal transducer and activator of transcription 4 (STAT4) and T-bet. The unique expression profiles of lncRNAs in different T cell subsets suggests that lncRNAs have critical roles in Th cell function during homeostasis and disease.

## 3. T Helper Cell Differentiation

To ensure T cells are capable of responding to pathogens, but do not generate an inappropriate response to antigens derived from the host, CD4^+^ T cells undergo a selection process in the thymus. Approximately 5% of CD4^+^ T cells leave the thymus and enter the blood stream [16]. As these newly developed cells have yet to encounter the target antigen, these are referred to as naïve CD4^+^ T cells. To become activated, the naïve CD4^+^ T cells must receive two signals. Signal 1 comprises the binding of the T cell antigen receptor (TCR) with antigen complexed to the major histocompatibility complex (MHCII) on an antigen-presenting cell (APC). Signal 2, also known as co stimulation, involves the binding of CD80 and CD86 on an APC to CD28 on the CD4^+^ T cell. The combination of signal 1 and 2 generates a cascading response which induces transcription factors such as nuclear factor of activated T cells (NFAT) and nuclear factor kappa light chain enhancer of activated B cells (NF-kB), thereby stimulating transcription of IL-2, which ultimately results in CD4^+^ T cell proliferation [17]. The T cell activation process is tremendously complex and alters the expression of thousands of differentially expressed genes. Given that lncRNAs have previously been implicated in gene regulation and that T cell activation drastically alters gene expression, this lends weight to the argument that lncRNAs should be studied in the context of T helper cells.

Following activation, T cells differentiate into effector T cells when provided with a third signal—cytokines. For example, in the case of Th1 induction, when activated T cells are exposed to IL-12, this stimulates high expression of key transcription factors such as T-bet and STAT4, which results in a Th1 phenotype. Contrastingly, when activated T cells are stimulated with IL-4 this generates high expression of STAT6 and GATA3 and a Th2 cell [17]. Indeed, cytokines and transcription factors are essential for effector CD4^+^ T cell differentiation, yet there is emerging evidence to suggest that lncRNAs also play a vital role in the differentiation processes (Figure 1). In addition, several lncRNAs have been implicated in other key aspects of CD4^+^ T cell function for example the T helper cell activation process. These lncRNAs will be discussed based on whether they affect genes in their immediate vicinity or those located further away on the chromosome or other chromosomes.

## 4. LncRNAs Affecting Immediately Neighbouring Genes through *In Cis* Interactions

One example of a functional lncRNA in CD4^+^ T cell differentiation is seen with the intergenic lncRNA (linc) linc-MAF-4. Linc-MAF-4 was first identified by RNA-seq data analysing human T cell subsets. It was found to be a Th1 specific lncRNA which represses MAF (avian musculoaponeurotic fibrosarcoma oncogene homolog) expression (predominantly a Th2 transcription factor). This repression is thought to occur through linc-MAF-4 interactions with chromatin modifiers enhancer of zeste homolog 2 (EZH2) and lysine-specific histone demethylase 1A (LSD1), which deposit H3K27me3 marks in the promotor region of MAF, thus silencing MAF expression. Subsequent linc-MAF-4 knock down in human peripheral blood mononuclear cells (PBMCs) skews CD4^+^ T cell differentiation towards a Th2 phenotype [18]. Supporting the data set, when naïve human CD4^+^ T cells are transfected with synthetic linc-MAF-4 this promotes Th1 differentiation and represses Th2 differentiation [19]. Collectively, this data demonstrates the key role of linc-MAF-4 in permitting a Th1 phenotype.

Another lncRNA that has been shown to be important in Th1 cells is *ne*ttoie *S*almonella pas *T*heiler’s (cleanup Salmonella not Theiler’s) (NeST), also known as IFNG-AS1 and originally TMEVPG1. NeST was initially identified as a candidate genetic factor for controlling Theiler’s virus persistence in the central nervous system [20]. This study noted that NeST and *Ifng* are neighbouring genes that are transcribed on opposite DNA strands in both mice and humans. Subsequent human and mouse CD4^+^ T cell in vitro polarisation experiments found that IFNγ and NeST were most highly expressed in Th1 cells. Interestingly, in vitro activated Th1 cells isolated from D011.10.Stat4^−/−^ and DO11.10.Tbx21^−/−^ mice showed diminished levels of both IFNγ and NeST. Moreover, subsequent siRNA knock down of NeST decreased expression of IFNγ [21], which indicated that NeST has an activating effect on IFNγ. To further characterise this finding, immunoprecipitation and qPCR experiments revealed that NeST interacts with WDR5, which is part of a complex of proteins that facilitates H3K4 methylation marks [22]. This study went on to generate in vivo data which revealed that in a mouse model of sepsis, mice that transgenically overexpress NeST exhibit greater expression of IFNγ and an increase in the proportion of H3K4me3 marks at the *IFN*γ locus [22]. Additional analysis has begun to characterise the mechanism by which NeST is regulated. Interestingly, T-bet was shown to bind to the NeST promoter and distal enhancers in Th1 cells, with NF-kB and Ets1 also shown to be important in this process [23], as formerly reviewed by [24]. Consistent with previous studies of NeST function, analysis of Hashimoto’s thyroiditis (HT) patients revealed that HT patients have an increased number of circulating Th1 cells with NeST, Tbet and IFNγ mRNA levels also upregulated. Most notably, NeST expression was positively correlated with the number of circulating Th1 cells, Tbet and IFNγ expression, providing an additional link between NeST and IFNγ [25]. Together, these studies clearly highlight the important role of NeST in Th1 cells and IFNγ expression.

Genome-wide analysis of primary human CD4^+^ subsets identified a Th2 specific lncRNA GATA3-AS1 [26]. GATA3-AS1 is known as a bidirectional promoter lncRNA, as it originates from the same promoter as GATA3, and they are transcribed in opposite directions. To understand the role of GATA3-AS1 in Th2 cells, the lncRNA was knocked down by siRNA in total human PBMCs stimulated under Th2 polarising conditions. Knock down led to a reduction of the signature Th2 polarised genes GATA3, IL-13 and IL-5, at both mRNA and protein levels [27]. Additionally, chromatin immunoprecipitation (ChIP) and PCR analysis in GATA3-AS1 knocked down cells revealed significantly reduced levels of the activating epigenetic marks H3K4me2/3 and H3K27ac across both the GATA3 and GATA3-AS1 genomic locus. To uncover the mechanism by which GATA3-AS1 alters chromatin marks further ChIP and DNA-RNA immunoprecipitation (DRIP) analysis was required. This demonstrated GATA3-AS1 RNA forms an R loop with the DNA in a central intron of GATA3-AS1 and that GATA3-AS1 binds to an essential component of the chromatin modifying MLL H3k4 methyltransferase complex. Although GATA3-AS1 could not induce a Th2 phenotype in the absence of factors such as MAF, these findings provide strong evidence that the lncRNA GATA3-AS1 plays an essential role in modifying the chromatin landscape of GATA3 and GATA3-AS1, in addition to regulating the expression of GATA3 and Th2 effector cytokines IL-5 and IL-13 [27].

Not only do lncRNAs play a role in Th1 and Th2 differentiation, there is evidence to support lncRNA function in Th17 differentiation. Microarray data from PBMCs of multiple sclerosis (MS) patients revealed that DNA-damage inducible transcript 4 (DDIT4) and the lncRNA DDIT4 (lncDDIT4) are upregulated in Th17 cells compared to other T cell subsets [28]. DDIT4 is a cytoplasmic protein which is upregulated upon DNA damage and is known to inhibit mTORC1 activity [29]. The mTOR pathway is crucial for a plethora of functions such as proliferation and has been implicated in positive regulation of Th17 differentiation [30]. lncDDIT4 is genomically located downstream of DDIT4, which suggests a potential regulatory role of lncDDIT4. This effect is seen when lncDDIT4 is knocked down in naive human CD4^+^ T cells causing decreased DDIT4 expression, and when the knocked down cells were stimulated under polarising conditions, this increased DDIT/mTOR signalling and bolstered Th17 cell numbers. Conversely, when lncDDIT4 was over expressed, DDIT4 expression was also enhanced. This effect resulted in lowered activation of the DDIT4/mTOR pathway meaning Th17 cell numbers were reduced. Importantly, alteration of lncDDIT4 levels did not affect Th1 or Th2 differentiation demonstrating a cell specific effect [28]. Future work to confirm and characterise the mechanistic interaction between lncDDIT4 and DDIT4 would be beneficial. Nevertheless, it is clear that lncDDIT4 plays a key role in Th17 differentiation.

## 5. LncRNAs Affecting Genes outside Their Immediate Vicinity

One of the first lncRNAs shown to be functional in T helper cells was the non-coding RNA nuclear repressor of NFAT (NRON). To uncover whether lncRNAs affected the function of the Ca^2+^ regulated transcription factor NFAT, an NFAT responsive luciferase assay was used to array a library of short hairpin RNAs (shRNAs) targeted towards 512 ncRNAs in Jurkat and HEK293 cells. Interestingly, shRNA targeting of NRON elevated activation of NFAT [31]. Subsequent in vitro binding experiments demonstrated NRON interactions with three members of the importin-beta superfamily, which are involved in transporting NFAT from the nucleus to the cytoplasm. However, later work in Jurkat and CD8^+^ T cells instead suggests that NRON binds to NFAT as part of an RNA-Protein scaffold complex comprising three NFAT kinases and the scaffold protein IQ motif containing GTPase activating protein (IQGAP). Yet, in further support of previous findings, knock down of NRON results in phosphorylation of NFAT, translocation to the nucleus, and higher levels of NFAT-regulated cytokines [32]. Additional investigations to study the suggested scaffold indicates that NFAT becomes uncoupled from the NRON scaffold by exploiting the transcription factor E26 transformation specific sequence 1 (Ets-1). Coimmunoprecipitation and ChIP analysis of WT and Ets-1KO Th cells showed that Ets-1 interacts with NFAT and synergistically activates the IL-2 promoter in the nucleus. In addition, upon Ca^2+^signalling, Ets-1 moves to the cytoplasm and competes with NFAT for the binding of the NRON complex, this releases NFAT from the complex enabling nuclear entry [33]. Despite the debate of the exact mechanism of NRON these studies provide compelling evidence for the crucial role of the lncRNA NRON in T helper activation.

Another example of a lncRNA affecting gene expression across the transcriptome is the lnc-epidermal growth factor receptor (EGFR). The role of lnc-EGFR in Tregs was discovered in a hepatocellular carcinoma (HCC) study in which high throughput screening examined links between lncRNAs and mRNAs in HCC patient samples [34]. LncEGFR was discovered to be highly expressed in Tregs and correlate positively with Foxp3 expression, but negatively with IFNγ. Moreover, RNA immunoprecipitation (RIP) analysis revealed that lnc-EGFR interacts with EGFR. Interestingly, overexpression and knock down of lnc-EGFR suggested that lnc-EGFR prevents ubiquitination of EGFR by blocking the effects of c-CBL. This study goes on to show binding sites for NFAT and AP1 are in the promoters of lnc-EGFR, EGFR and FOXP3, and expression of all three increases upon NFAT and AP1 signalling. The proposed model suggests that lnc-EGFR acts in a forward feedback loop lnc-EGFR-EGFR-NFAT/AP1-lnc-EGFR which promotes Treg differentiation and clearly showcases the mechanism of lnc-EGFR.

X inactive-specific transcript (Xist) is a well-studied lncRNA, predominantly known for its function in X chromosome inactivation. In-depth analysis describes it as silencing of the majority of genes along the inactive X chromosome by recruiting epigenetic modifiers that deposit repressive chromatin marks [35,36,37]. Fascinatingly, RNA fluorescence in situ hybridisation (FISH) analysis of both human and murine naïve female CD4^+^ T cells shows dispersed Xist localisation, similar to that observed in male T cells. Upon in vitro T cell activation, Xist returns to the Xi in female cells; however, levels of repressive chromatin marks such as H3K27me3 are notably reduced. Knock down experiments demonstrate that the return of XIST to the Xi upon T cell activation is YY1- and hnRNPU-dependant. As a result of lower repressive chromatin marks, the inactive X chromosome becomes partially reactivated; thus, bi allelic expression of X linked immune genes CD40LG and CXCR3 occurs in a small percentage of cells [38]. As CD40L and CXCR3 are strongly associated with the autoimmune disorder systemic lupus erythematosus (SLE), this inspired further work to characterise potential links between XIST localisation and the autoimmune disease. Interestingly, splenic T cells isolated from SLE patients and late stage NZB/W F1 female mice (spontaneous lupus-like disease mice with strong female bias), show dispersed XIST localisation. Consequently, gene expression analysis of SLE patients splenic T cells showed an abnormal upregulation of transcription along X chromosome and altered expression of XIST RNA binding proteins [39]. In further support of the importance of correct XIST localisation in T cells, when Cip1-interacting zinc finger protein (CIZ1) (a protein recruited to the Xi by XIST) is knocked out, Ciz1 null mice develop fully penetrant female-specific lymphoproliferative disorder, and Xist localisation is disrupted in splenic T cells in the absence of CIZ1 [40]. Collectively, these studies suggest that XIST localisation at the Xi could be a crucial factor in maintaining dosage compensation of X linked genes in T cells. Moreover, this provides a potential argument that XIST may be a contributing factor in mammalian females’ ability to better combat pathogens comparative to males and their increased predisposition to autoimmune disorders.

## 6. Other lncRNAs

Remarkably, the literature describes conflicting results for the role the lncRNA metastasis associated lung adenocarcinoma transcript 1 (MALAT1) in the context of T helper cells. A recent in vivo study assessed the role of MALAT1 in CD4^+^ T cell responses to acute infection with lymphocytic choriomeningitis virus (LCMV) [41]. Flow cytometry analysis revealed no significant difference in the proportion of different peripheral CD4^+^ T cell populations between MALAT1 KO and WT mice after 8 days of infection. Contrastingly, siRNA mediated knock down of MALAT1 in primary naïve T cells pushes activated T cells towards a Th1/Th17 phenotype and inhibits Treg differentiation in vitro [42]. To determine if MALAT1 is truly non-functional in a T helper context, further investigation is required.

Another lncRNA nuclear-enriched abundant transcript 1 (NEAT1) has been implicated in both Th2 and Th17 differentiation. A recent study examining PBMCs from rheumatoid arthritis patients indicated that NEAT1 was elevated in Th17 cells and demonstrated that knockdown of NEAT1 inhibited Th17 differentiation. It is thought that NEAT1 regulates this differentiation process through interactions with STAT3 [43]. However, another recent study suggests that NEAT1 promotes Th2 differentiation [44]. RIP and ChIP analysis revealed that NEAT1 binds to EZH2, which is recruited to the ITCH promoter repressing ITCH, in addition expression of STAT6 increased along with IL-4, IL-5 and IL-13. Together, these studies suggest that NEAT1 may have multiple roles in T helper cell differentiation.

Other lncRNAs have been implicated in CD4^+^ T cell function in the context of human immunodeficiency virus (HIV) infection. Interestingly, deep sequencing of HIV-1 infected human CD4^+^ T cells and SUP-T1 cells (a human T cell line) revealed hundreds of differentially expressed lncRNAs over the course of HIV-1 infection [45,46,47]. A later study conducted meta-analysis of two RNA-seq experiments of HIV-1 infected and non-infected T cells. This identified 3 lncRNAs which had over 2-fold altered expression upon HIV-1 infection. The chosen lncRNA for further study, lncRNA lnc173, was not shown to affect HIV-1 replication in T cells. However, lnc173 KO Jurkat T cells expressed higher mRNA levels of *IFNG*, *CCL3* and *CXCL8*. In addition, increased protein expression of IFNγ was observed in some of the lnc173 KO clones screened. This indicates that lnc173 may have a potential role in the regulation of these cytokines [48]. Interestingly, transcript levels of both MALAT1 and NEAT1 have been shown to be upregulated in PBMCs of HIV-1-infected patients, and, notably, expression of NEAT1 correlated with CD4^+^ T cell counts, indicating potential function of these lncRNA during HIV-1 infection [49]. Interestingly, HIV-1 replication is higher in NEAT1 KO Jurkat cells compared to WT, suggesting a potential anti-viral effect [50]. Contrastingly, HIV-1 replication is lower in MALAT1 KO Jurkat cells, and MALAT1 is thought to enhance infection by detaching EZH2 from binding the HIV-1 LTR promoter [51]. The data from these studies warrant further research into the role of lncRNAs in the context of HIV-1 infection.

Several other lncRNAs are known to be fundamental in CD4^+^ T cell functions (Table 1), some of which are reviewed by [24]. Examples of these include a Th2 specific lncRNA cluster Th2LCR which regulates differentiation by affecting IL-4, IL-5 and IL-13 expression [52]. A novel lncRNA Flatr is thought to act in the Treg differentiation pathway in mice [53]. Additionally, the lncRNA highly upregulated in liver cancer (HULC) is able to reduce P18 expression which impairs Treg differentiation [54]. Together these studies provide compelling support for the essential role of lncRNAs in CD4^+^ T cell differentiation, thus, enabling the immune system to elicit targeted responses to pathogens.

## 7. Future Challenges and Solutions

Analysis of lncRNAs in CD4^+^ T cells presents, on the whole, compelling evidence that the non-coding genome accounts for some of the complexity observed in mammalian immunity. Various knock down and overexpression experiments have demonstrated their crucial role in the differentiation of Th1, Th2, T17 and Treg cells. Additionally, several lncRNAs such as IFNG-AS1 have been implicated in CD4^+^ T cell mediated pathologies such as myasthenia gravis (MG) [66], which offers further support for the physiological role of lncRNAs.

In the case of CD4^+^ T cells, the implicated functional lncRNAs do not appear to have a bias in their mode of action (local or distal gene regulation). It is, therefore, logical to take experimental approaches that lack prejudice, such as that suggested by Kopp and colleagues to uncover other lncRNA functions in a T helper context [5]. This approach first suggests gaining an understanding of the transcript structure, followed by loss-of-function experiments and examination of local gene expression changes, which would initially help discriminate between local and distal function. To facilitate these discoveries, improved techniques which enable the study of lncRNA interactions with DNA, RNA and proteins may hold the key to deepening our understanding of the mechanistic relevance of lncRNAs in CD4^+^ T cells. One example of such a technique is RNA antisense purification (RAP), developed by the Guttman lab. RAP can be used in combination with DNA/RNA sequencing or mass spectrometry (MS) to identify lncRNA interaction partners. These RNA-centric approaches offer many advantages, for example with regard to identifying new RNA–protein interactions. Typical in vitro RNA purification methods involve the generation of cellular extracts and subsequent incubation with synthetic RNA bait. However, these in vitro methods often give rise to false positive results and identify binding which occurs in the context of cellular contents in solution, meaning that these identified proteins do not truly represent the interactions taking place inside a cell. To overcome these barriers, RAP-MS uses ultra violet (UV) light to directly crosslink lncRNA protein interactions in vivo. Subsequent lncRNA pull down using long antisense 5′ biotinylated probes enable purification of these lncRNAs under highly reducing and denaturing conditions. This method more accurately represents true lncRNA protein interactions, and has been used to identify XIST interaction partners [67]. Gold standard pull down methods such as cross-linking immunoprecipitation (CLIP) also employs the use of UV light to establish in vivo covalent crosslinking. However, as this approach focusses on protein pull downs it limits the ability of identifying new RNA binding proteins (RBPs). Consequently, further development of RAP techniques or similar RNA centric pull down strategies such as capture hybridisation analysis of RNA targets (CHART) [68], and chromatin isolation by RNA purification (ChIRP) [69] are crucial for expanding our insight into the role of lncRNAs in CD4^+^ T cells and their mechanism.

Interestingly, the functional relevance of only a handful of lncRNAs has been validated in vivo. Given the relatively small body of literature, to help broaden our understanding of the roles of lncRNAs a multitude of infection and chronic inflammation studies beyond standard models such as LCMV infection may be necessary to reveal the functional contributions of lncRNAs, and would crucially demonstrate their physiological relevance. However, many lncRNAs have a low degree of evolutionary conservation which poses some difficulties, as what is observed in mice may not be reflective of that seen in man. This poor level of conservation may be a contributing factor in the lack of mouse models for lncRNAs in the context of T helper cell immunity. Nevertheless, it remains a problem which needs to be addressed. There are approaches available which may help circumvent these issues with advances in animal modelling such as “humanised mice” perhaps being of interest for this field. Humanised mice are immunodeficient mice (e.g., *IL2rg^null^*) which are engrafted with human haematopoietic stem cells or tissues. These cells and tissues enable the development of a functional human immune system with their immune responses being more representative of those seen in humans comparative to other mouse models as reviewed by [70]. Conceivably, the use of this technique in combination with CRISPR/Cas9 genome editing technology [71] to modify human lncRNA expression levels provides a possible solution this problem. Progress has been made in the development of new CRISPR/Cas9 technologies, which enables reversible control of gene expression. These methods exploit the use of nuclease-dead mutants of Cas9 (dCas9) and can inhibit transcription by CRISPR interference (CRISPRi) or increase transcription levels using CRISPR activation (CRISPRa) [72]. These technologies are uniquely suitable for the study of lncRNAs as they specifically regulate transcription and can control the levels of transcript by several orders of magnitude [73]. This is particularly beneficial, as, in the case of many lncRNAS, transcription rather than the RNA itself is important, and in many others, the nascent transcript is the functional entity [74]. The combination of a humanised mouse model with these CRISPR technologies to study lncRNAs has the major advantage of providing a more accurate representation of the function of human lncRNAs in a manipulatable in vivo context. Using these tools would be of particular benefit in the testing of lncRNA therapeutics.

As many lncRNAs are expressed in a cell specific manner, this makes them ideal candidates for therapeutic intervention. Several methods of targeting lncRNAs are beginning to be developed, including siRNA-mediated degradation antisense oligonucleotides (ASOs) that have been chemically modified, sterically blocking lncRNA promoters and in addition, steric inhibition of RNA protein interactions via ASO or small molecules [75]. siRNA/ASO technology has been successfully used to knockdown the expression of lncRNAs in cell lines; however, in a recurring theme, only a few have been tested in vivo. One example of successful in vivo analysis of ASO effectiveness is seen with systemic knockdown of MALAT1 using subcutaneous delivery of ASO in a mouse mammary tumour virus (MMTV) PyMT model; this caused primary tumours to differentiate and there was a significant reduction in metastases [76]. The FDA has recently approved the use of antisense drugs for Duchene muscular dystrophy, which has led to the development of multiple pre-clinical models reviewed by [75]. An interesting pre-clinical technique to study lncRNAs that are restricted to humans is patient-derived xenograft models (PDXs), in which PDX tumours are implanted into nude mice. The lncRNAs HOTAIR and SAMMSON have been targeted by ASO/siRNA in these models [77,78]. An accumulating body of evidence has implicated lncRNAs in a plethora of complex human diseases, ranging from multiple cancers to autoimmune diseases [79]. As CD4^+^ T cells play a critical role in the pathogenesis of several autoimmune disorders such as multiple sclerosis (MS), rheumatoid arthritis (RA) and SLE, further understanding the mechanistic role of lncRNAs in CD4^+^ mediated pathology may unveil an untapped treasure trove of drug targets, biomarkers and enhance our knowledge of the immune system.

Other classes of ncRNAs have also been shown to have crucial roles in T helper cell biology, one example of which is circular RNAs (circRNAs) [80]. These are a distinct class of ncRNAs which are characterised by their covalently linked loop structure and lack of both 5′ and 3′ ends [80,81]. Examples of their roles in T helper cell function range from regulation the Akt (Protein Kinase B) signalling pathway, to links with CD4^+^ mediated pathologies [82,83,84]. However, it appears that little work has focused on the potential functions of circRNAs in T helper cell differentiation. Studying their potential functions in CD4^+^ T cells is necessary to gain further insight into the role of the non-coding transcriptome in adaptive immunity.

In conclusion, the study of lncRNAs in the context of T helper responses is an exciting area of research for deepening our understanding of the immune system. With the rise of innovative techniques such as humanised mouse models, these provide ideal platforms for further probing the mechanistic relevance of lncRNAs, which, once it is better understood, has promising implications for the development of potential treatments in CD4^+^-mediated pathology.

## Figures and Tables

**Figure 1 ncrna-05-00043-f001:**
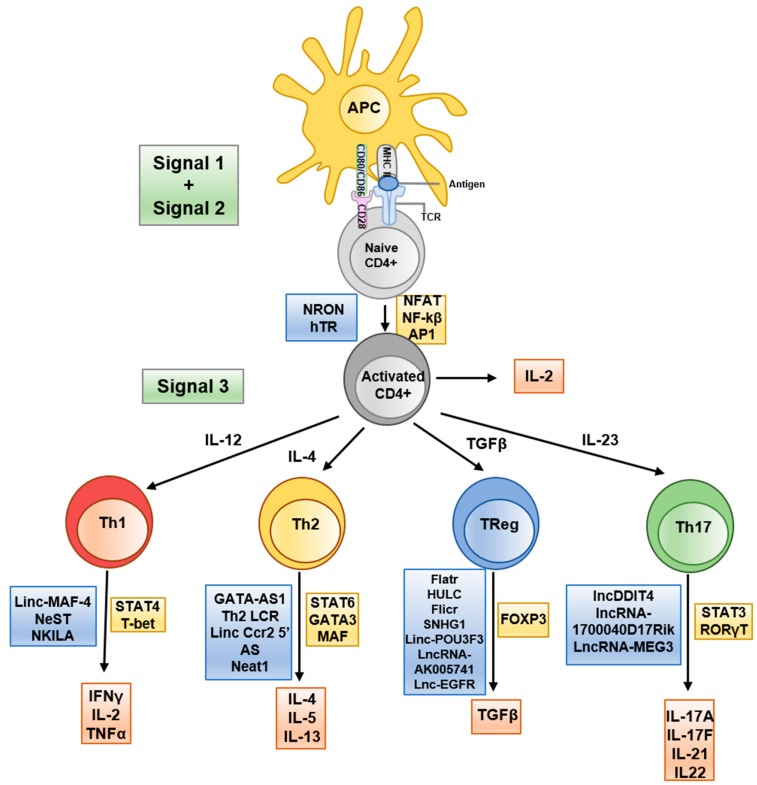
Schematic representation of T helper differentiation and associated lncRNA. Key secreted cytokines (red boxes), transcribed transcription factors (yellow boxes) and implicated lncRNAs (blue boxes) are indicated at the appropriate transitional stages.

**Table 1 ncrna-05-00043-t001:** lncRNAs known to influence CD4^+^ T cell function.

lncRNA	Subset	Target	Mechanism	CD4^+^ In Vivo?	References
*LncRNAs affecting immediately neighbouring genes*					
linc-MAF-4.	Th1	MAF	Interacts with EZH2 and LSD1 to epigenetically repress MAF.	No	[18,19]
NeST	Th1	IFNγ	H3K4-methylation at IFNG promoter aids transcription	Yes	[20,22,25]
GATA-AS1	Th2	GATA3, IL-13, IL-5	Epigenetic modification of GATA3 chromatin landscape	No	[26,27,52]
Th2 LCR (Cluster of lncRNA)	Th2	IL-4, IL-13, IL-15	Histone acetylation, methylation and demethylation in the Th2 cytokine locus.	Yes	[52,55,56]
lncRNA-1700040D17Rik	Th17	RORγt	Potential *local* regulation	No	[57]
lncDDIT4	Th17	DDIT4/mTOR	Potential *in local* regulation	No	[28]
Flicr	Tregs	Foxp3	Negative regulator modifies FOXP3 chromatin access	Yes	[58]
*LncRNAs affecting gene expression outside their immediate vicinity*					
Lnc-EGFR	Treg	EGFR receptor	Binds EGFR directly, stabilising it and augmenting activation enabling EGFR expression	Yes	[34]
NRON	Activated CD4^+^	NFAT	Sequesters NFAT in the cytoplasm	No	[31,32]
HULC	Treg	P18	HULC binds P18 potential direct inhibition	No	[54].
Linc-POU3F3	Treg	TGF-B	Binds TGF-B increasing phosphorylation of SMAD 2/3	No	[59]
SNHG1	Treg	miR-488	Facilitates Tregs by potential interactions with miR-488 and IDO	No	[60]
NKILA	Th1	ACID	Binds and inhibits NFkB	No	[61]
LncRNA-MEG3	Th17	RORyt	Acts as a miRNA sponge	No	[62,63]
Xist	CD4^+^ Cells	X chromosome genes	Silencing immune linked genes along the inactive X chromosome	No	[38,39]
Yet to be characterised					
Flatr	Treg	Foxp3	Unknown	Yes	[53]
Linc-Ccr2-5′AS	Th2	CCR1, CCR3, CCR2, CCR5	Unknown	Yes	[15]
lncRNA-AK005641	Treg	Differentiation	Unknown	No	[64]
hTR	CD4^+^	Apoptosis	Unknown	No	[65]

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
