# Peer review of "Long Non-Coding RNA Function in CD4+ T Cells: What We Know and What Next?"

_ncrna, 2019, doi:10.3390/ncrna5030043_

Round 1
Reviewer 1 Report
The review manuscript by West and Lagos focuses on the role of lncRNAs in CD4+ T cells. This is a very interesting and high impact field and the role of this class of transcripts in the immune response is understudied. However, there are a number of issues that have to be addressed before the review article is ready for publication.
1) The authors completely ignore the HIV literature. CD4+ T cells have been extensively studied due to their involvement in HIV/AIDS and several lncRNAs have been discovered to affect the function of CD4+ T cells in this context. The authors have discussed lncRNAs induced in CD4+ T cells in response to other viral infections, appropriately. They must also cover the studies on the role of lncRNAs in CD4+ T cells in the context of HIV infection. There are high throughput studies (Trypsteen W1,Mohammadi P2,Van Hecke C1,Mestdagh P3,Lefever S3,Saeys Y4,5,De Bleser P4,6,Vandesompele J3,Ciuffi A2,Vandekerckhove L1,De Spiegelaere W1,7.Differential expression of lncRNAs during the HIV replication cycle: an underestimated layer in the HIV-host interplay. Sci Rep. 2016 Oct 26;6:36111. doi: 10.1038/srep36111) and low throughput studies from multiple labs. These studies should be included in the review.
2) Line 30: this is factually incorrect. The protein-coding “genes” do not cover 2% of the genome, the figure is closer to 37%. Rather, the fully spliced, protein-coding transcript isoforms of these genes comprise just above 2% of the genome. This is an important distinction and must be corrected.
3)Line 39:Including siRNAs in the category of small non-coding RNAs is somewhat inappropriate. In higher eukaryotes, which is the focus of this review, there are no endogenous siRNAs. The authors can list snRNAs, snoRNAs and tRNAs as other, legitimate examples of small non-coding RNAs.
4)Line 43: the sentence starting with “Typically”. Typically means the predominant mode of what is being described. The predominant mode of lncRNAs is localization to the nucleus, not both compartments. The rest of the sentence indeed reflects the properties of a “typical” lncRNA.
5)lines 49-52: the definition of cis and trans is rather outdated since there are many exceptions and nuances. For example, lncRNA FIRRE: it contacts regions located far away on the chromosome due to the three dimensional structure of chromatin. For such lncRNAs, the nascent transcript can affect far away loci, and assigning a “cis” or “trans” designation is rather difficult in such cases. The authors need to indicate such nuances in the review and in general progress beyond these outdated designations.
6) Line 150: The formal biotype for such RNAs according to HGNC and GENCODE is “Bidirectional promoter lncRNA”, not “divergent lncRNA”. One can say that the lncRNAs originating from bidirectional promoters are a special case of divergent configuration, but it’s important to use the official designation in a review article.
7) Line 150 again: also, it is not "located" in the same promoter as stated in the text (line 150), it "originates" from the same promoter or transcribed from the same promoter.
8) Line 194: The authors need to clarify what NZB/W mice are. They can’t expect the readers to automatically know this.
9) Line 235: TO clarify the "in trans" mechanism it is important to indicate that the gene giving rise to lnc-EGFR is not located close to any of its targets.
10)Line 242: please clarify what is meant by peripheral cells. Is it PBMCs? If so, the full name should be used to remove ambiguity.
11) In the last section of the review which includes the discussion of the technologies, it is important to also include the CRISPRi and CRISPRa technologies, which are uniquely suitable for the study of lncRNAs as these techniques regulate transcription. Since in the case of many lncRNAs the act of transcription, rather than the RNA itself is important, and in many others, the nascent transcript is the functional entity, such technologies are highly suitable for use in studying the function of lncRNAs.
p { margin-bottom: 0.1in; line-height: 120%; }a:link { }
Author Response
We thank the Reviewer for the useful suggestions. We have now addressed them as detailed below:
1) The authors completely ignore the HIV literature. CD4+ T cells have been extensively studied due to their involvement in HIV/AIDS and several lncRNAs have been discovered to affect the function of CD4+ T cells in this context. The authors have discussed lncRNAs induced in CD4+ T cells in response to other viral infections, appropriately. They must also cover the studies on the role of lncRNAs in CD4+ T cells in the context of HIV infection. There are high throughput studies (Trypsteen W1,Mohammadi P2,Van Hecke C1,Mestdagh P3,Lefever S3,Saeys Y4,5,De Bleser P4,6,Vandesompele J3,Ciuffi A2,Vandekerckhove L1,De Spiegelaere W1,7.Differential expression of lncRNAs during the HIV replication cycle: an underestimated layer in the HIV-host interplay. Sci Rep. 2016 Oct 26;6:36111. doi: 10.1038/srep36111) and low throughput studies from multiple labs. These studies should be included in the review.
Response: We have now added a paragraph (lines 272-288) and the suggested references exploring the roles of lncRNA in the context of HIV infection.
2) Line 30: this is factually incorrect. The protein-coding “genes” do not cover 2% of the genome, the figure is closer to 37%. Rather, the fully spliced, protein-coding transcript isoforms of these genes comprise just above 2% of the genome. This is an important distinction and must be corrected.
Response: We have now clarified our statement as per reviewer’s suggestion (lines 29-30).
3)Line 39:Including siRNAs in the category of small non-coding RNAs is somewhat inappropriate. In higher eukaryotes, which is the focus of this review, there are no endogenous siRNAs. The authors can list snRNAs, snoRNAs and tRNAs as other, legitimate examples of small non-coding RNAs.
Response: We have now amended the text as per reviewer’s suggestion (line 40).
4)Line 43: the sentence starting with “Typically”. Typically means the predominant mode of what is being described. The predominant mode of lncRNAs is localization to the nucleus, not both compartments. The rest of the sentence indeed reflects the properties of a “typical” lncRNA.
Response: We have now amended the text as per reviewer’s suggestion.
5)lines 49-52: the definition of cis and trans is rather outdated since there are many exceptions and nuances. For example, lncRNA FIRRE: it contacts regions located far away on the chromosome due to the three dimensional structure of chromatin. For such lncRNAs, the nascent transcript can affect far away loci, and assigning a “cis” or “trans” designation is rather difficult in such cases. The authors need to indicate such nuances in the review and in general progress beyond these outdated designations.
Response: We fully agree and have simplified our terminology to distinguish between lncRNAs that have been shown to only affect expression of genes in their chromosomal vicinity versus lncRNAs that affect expression of distal genes. The text has been amended in throughout. The Table has also been modified accordingly.
6) Line 150: The formal biotype for such RNAs according to HGNC and GENCODE is “Bidirectional promoter lncRNA”, not “divergent lncRNA”. One can say that the lncRNAs originating from bidirectional promoters are a special case of divergent configuration, but it’s important to use the official designation in a review article.
Response: We have now amended the text as per reviewer’s suggestion (line 196-197).
7) Line 150 again: also, it is not "located" in the same promoter as stated in the text (line 150), it "originates" from the same promoter or transcribed from the same promoter.
Response: We have now amended the text as per reviewer’s suggestion (line 161).
8) Line 194: The authors need to clarify what NZB/W mice are. They can’t expect the readers to automatically know this.
Response: NZB/W mice are explained now in the text (line 240).
9) Line 235: TO clarify the "in trans" mechanism it is important to indicate that the gene giving rise to lnc-EGFR is not located close to any of its targets.
Response: We have now amended the text as per reviewer’s suggestion (line 215-217).
10)Line 242: please clarify what is meant by peripheral cells. Is it PBMCs? If so, the full name should be used to remove ambiguity.
Response: We have now amended the text as per reviewer’s suggestion (line 133).
11) In the last section of the review which includes the discussion of the technologies, it is important to also include the CRISPRi and CRISPRa technologies, which are uniquely suitable for the study of lncRNAs as these techniques regulate transcription. Since in the case of many lncRNAs the act of transcription, rather than the RNA itself is important, and in many others, the nascent transcript is the functional entity, such technologies are highly
suitable for use in studying the function of lncRNAs.
Response: We have now added a paragraph (lines 350-360) discussing CRISPRi and CRISPRa (lines 471-486).
Reviewer 2 Report
Authors present a very elegant and clear review on the role of lncRNAs for the activation and differentiation of CD4 + T cells. The manuscript contemplates the major lncRNAs involved in the Th1, Th2, Th17 and Treg profiles, exercising their function in cis or trans mode, which impacts in modulation of transcription factors nuclear translocation and induction of cytokines/chemokines expression. I strongly recommend the publication of the manuscript.
Author Response
We thank the Reviewer for the positive evaluation of our manuscript.
Reviewer 3 Report
West and Lagos provide a well-written review of the role lncRNAs play in T helper cells, specifically with regard to differentiation of T helper cells. The table they provide nicely reflects current understanding. While other recent reviews have discussed the roles of lnRNAs in T cells, none have focused on the differentiation of T helper subsets like this one. As such, this is a nice addition to the literature and should be accepted for publication.
Considerations:
1) The focus of this review is really on the role of the lncRNAs in T helper cell differentiation, as specifically mentioned on lines 111 and 112. Two of the discussed lncRNAs, however, didn't quite fit this predominant focus: NRON and Xist. The discussion of these lncRNAs, however, centered on forms of activation of T helper cells. Because this is also relevant it should be included, but it does not exactly fit the main focus of the review. The authors should consider changing some of the language in the paragraph of lines 105-112 to moderate the expectations of the reader.
2) Xist should also be included in the table.
3) In the futures section, it might be interesting to add some discussion about the potential roles of another class of ncRNAs, circRNAs, in T helper cell differentiation. There's already been quite a bit of work on circRNAs and their roles in T cell biology, but it appears that little (none to my knowledge) work has focused on potential functions of circRNAs in their differentiation.
Minor points:
"in to" should be "into" in several places in the manuscript.
Line 28 - add reference for 80% statement.
In many places the plural of lncRNA (lncRNAs) should be used: e.g. line 46
Line 53, confusingly written. Maybe add a comma after "eukaryotes"?
Line 92 "tolerant to self-antigen" is there a better term that can be used?
Line 103-4 "lends weight to argument of" re-write "lends weight to the argument that lncRNAs should be studied in the context of T helper cells" ...or something like that.
Line 117, please define "MAF"
Line 127, NeST is written as IFNG-AS1 in the figure, either replace this in the Fig. or write "IFNG-AS1 (also known as NeST)" in the text.
Line 129, "this study"
Line 176-7, check the grammar here.
Author Response
We thank the Reviewer for the helpful suggestions on our manuscript. These are addressed below:
Considerations:
1) The focus of this review is really on the role of the lncRNAs in T helper cell differentiation, as specifically mentioned on lines 111 and 112. Two of the discussed lncRNAs, however, didn't quite fit this predominant focus: NRON and Xist. The discussion of these lncRNAs, however, centered on forms of activation of T helper cells. Because this is also relevant it should be included, but it does not exactly fit the main focus of the review. The authors should consider changing some of the language in the paragraph of lines 105-112 to moderate the expectations of the reader.
Response: We have now clarified that our review focuses on the role of lncRNAs on differentiation, activation, and function of T helper cells.
2) Xist should also be included in the table.
Response: We have now included XIST in the table.
3) In the futures section, it might be interesting to add some discussion about the potential roles of another class of ncRNAs, circRNAs, in T helper cell differentiation. There's already been quite a bit of work on circRNAs and their roles in T cell biology, but it appears that little (none to my knowledge) work has focused on potential functions of circRNAs in their differentiation.
Response: We have now included a paragraph on circRNAs (lines 382-389).
Minor points:
"in to" should be "into" in several places in the manuscript.
Line 28 - add reference for 80% statement.
In many places the plural of lncRNA (lncRNAs) should be used: e.g. line 46
Line 53, confusingly written. Maybe add a comma after "eukaryotes"?
Line 92 "tolerant to self-antigen" is there a better term that can be used?
Line 103-4 "lends weight to argument of" re-write "lends weight to the argument that lncRNAs should be studied in the context of T helper cells" ...or something like that.
Line 117, please define "MAF"
Line 127, NeST is written as IFNG-AS1 in the figure, either replace this in the Fig. or write "IFNG-AS1 (also known as NeST)" in the text.
Line 129, "this study"
Line 176-7, check the grammar here.
Response: We have now checked and amended the text as per all the above suggestions from the reviewer.